# Recognition of Hand Gestures Based on EMG Signals with Deep and Double-Deep Q-Networks

**DOI:** 10.3390/s23083905

**Published:** 2023-04-12

**Authors:** Ángel Leonardo Valdivieso Caraguay, Juan Pablo Vásconez, Lorena Isabel Barona López, Marco E. Benalcázar

**Affiliations:** 1Artificial Intelligence and Computer Vision Research Lab, Escuela Politécnica Nacional, Quito 170517, Ecuador; angel.valdivieso@epn.edu.ec (Á.L.V.C.); lorena.barona@epn.edu.ec (L.I.B.L.); 2Faculty of Engineering, Universidad Andres Bello, Santiago, Chile; juan.vasconez@unab.cl

**Keywords:** hand gesture recognition, electromyography, reinforcement learning, Deep Q-Network, Double-Deep Q-Network

## Abstract

In recent years, hand gesture recognition (HGR) technologies that use electromyography (EMG) signals have been of considerable interest in developing human–machine interfaces. Most state-of-the-art HGR approaches are based mainly on supervised machine learning (ML). However, the use of reinforcement learning (RL) techniques to classify EMGs is still a new and open research topic. Methods based on RL have some advantages such as promising classification performance and online learning from the user’s experience. In this work, we propose a user-specific HGR system based on an RL-based agent that learns to characterize EMG signals from five different hand gestures using Deep Q-network (DQN) and Double-Deep Q-Network (Double-DQN) algorithms. Both methods use a feed-forward artificial neural network (ANN) for the representation of the agent policy. We also performed additional tests by adding a long–short-term memory (LSTM) layer to the ANN to analyze and compare its performance. We performed experiments using training, validation, and test sets from our public dataset, EMG-EPN-612. The final accuracy results demonstrate that the best model was DQN without LSTM, obtaining classification and recognition accuracies of up to 90.37%±10.7% and 82.52%±10.9%, respectively. The results obtained in this work demonstrate that RL methods such as DQN and Double-DQN can obtain promising results for classification and recognition problems based on EMG signals.

## 1. Introduction

Hand gesture recognition (HGR) systems have been one of the most studied research topics in recent years due to their various applications in the area of human–machine interaction. This is partly due to all the possible benefits of these systems, such as their adaptability, versatility, ease of use, and the possible benefits of controlling a machine with hand gestures and movements. It is particularly interesting to develop systems that are capable of giving orders with gestures made in the air, which can replace or complement traditional systems that require touching a screen, a button, or a joystick [1,2]. The applications of HGR systems include communication systems with chatbots, natural language processing, prosthesis development, virtual reality, augmented reality, human–robot interaction, internet browsing, control and navigation of assistance vehicles, robotics applications in medicine, and video game control, among others [1,2,3,4]. However, designing HGR systems that are able to determine online and precisely when a certain gesture is performed is a challenging problem. This is, in part, due to the fact that the signals and information obtained from each user present a certain variability, since, for example, each user performs the same gestures in a slightly different way. Therefore, even when users perform the same gesture, the obtained signals differ. Likewise, it may be the case that there are similarities in the measurements obtained between different hand gestures, which may result in an erroneous recognition of the user’s intentions [2,5,6,7].

In general, the HGR systems that have been reported in the literature use vision-based methods, such as RGB cameras, thermal cameras, infrared cameras, depth cameras, stereo cameras, leap motion systems, Kinect, LiDAR sensors, and colored gloves [2,8,9,10,11,12]. HGR systems can also use different sensors, such as gloves embedded with flex sensors, ultrasound sensors, muscle vibration sensors, Wi-Fi signal information, inertial measurement units (IMU), and electromyography (EMG) electrodes, to analyze finger, hand, and arm movements [2,13,14,15,16,17,18,19]. However, it is crucial to note that occlusion, the distance between the hand and the sensor, environmental noise, and lighting issues can all have a negative impact on how well vision-based systems operate. Therefore, sensor-based HGR systems such as EMG, ultrasonic, or IMU-based techniques are frequently chosen for various HGR applications [17,18,19,20,21,22]. While IMU and VI signals have proven to be very useful for recognizing dynamic gestures (gestures that depend on arm movement), EMG signals are still the most widely used for recognition of static hand gestures (only movement related to the fingers and joints of the hand). This is because it has been shown that muscle activity can be characterized using EMGs for HGR applications. This has even led to the development of commercial sensors based on EMG signals, such as the Myo armband and G-force [15].

Creating a mathematical model based on EMG signals is challenging, since muscle activity can be seen as a stochastic process. However, approaches that model EMGs are not used for HGR applications because calculating or estimating parameters is complex [23,24]. In order to address these challenges, several methods have been investigated to identify EMG signals for HGR systems [1], such as supervised statistical machine learning (ML) methods such as Bayesian inference, maximum likelihood, support vector machine (SVM), decision trees, k-nearest neighbors (K-NN), hidden Markov models, the AdaBoost learning algorithms, and Haarlet methods. Deep learning (DL) methods have also been investigated as novel and promising techniques for EMG signal classification for HGR systems, such as artificial neural networks (ANNs), convolutional neural networks (CNNs), recurrent neural networks (RNNs), long–short-term memory networks (LSTMs), and generative adversarial networks (GANs) [1,2,11,21,25,26,27,28,29]. Models for HGR applications that are based on ML or DL learn patterns from a dataset to make inferences. In HGR applications based on EMGs, those models require a fully labeled dataset of EMG information with all the labels of the gesture categories to be trained. One of the problems with traditional ML or DL techniques is that if they require the addition of new data for training, the entire dataset must be processed again, which makes online training difficult. Another possibility is to use reinforcement learning (RL) algorithms, which do not require labels. These algorithms are based on an agent that learns an optimal policy by using experiences (rewards and penalties) to perform a specific activity within a particular environment. RL-based algorithms try to maximize the reward accumulated during each episode. This can bring benefits to the development of HGR systems, since it is easier to integrate information that allows an agent to learn online as a user interacts with the system, as well as to reduce the signal variability problems using autocalibration [30]. According to [31], RL-based methods have been previously used successfully in learning from biological data applications. For example, RL can be applied to brain–body interface applications using electroencephalogram (EEG), electrocardiogram (ECG), and electromyogram (EMG) signals. However, to date, most applications that work on biological data with RL are based on finding patterns, decoding, or anomaly detection related to the functions of the human brain, for example, to controll a robotic arm or prosthesis or map neural activity using actor–critic architectures [31,32,33]. However, the problems related to HGR and RL systems are just beginning to be investigated; therefore, evaluating their use, performance, and benefits is key to developing this research area.

In the literature, some works have been found that use RL techniques for HGR applications and/or arm movement classification. However, only a few of those approaches use EMG signals to acquire information about the hand or arm gestures to characterize them using RL methods. In [34], an HGR approach capable of analyzing EMG using Deep Q-Networks (DQNs) and dueling EMG was developed. The authors obtained the EMG information from the UCI dataset, which contains EMGs related to six different hand gestures (grasp, point, hook, palmar, spherical, and lateral). There were a total of 2700 EMG samples in the dataset, and data augmentation (adding noise and random flip) was used to increase the data size up to 10,000 samples. The features used to classify the EMG information were RMS, zero crossing times, slope sign change, and mean absolute value. Those features were obtained using a sliding window approach, then concatenated and resized into a fixed-size matrix that was processed by a CNN to obtain the classification results. Using the RL approach, the author achieved a classification accuracy of 87.5%, which represented an improvement relative to the ML approach, which achieved an accuracy of 78.85%. In [35], RL-based models were developed to classify hand, finger, and arm movements based on EMGs. To recognize elbow angle, EMGs were recorded with different weights (efforts) for different elbow positions; to recognize key typing classes, the EMG data of four binding keys were used; and for hand movement classification, the EMG data corresponding to six different hand movements (open, closed, supination, pronation, flexion, and extension) were used. The dataset was composed of 10 users’ information, in which 144 samples for training and 95 for testing for each of the subjects were recorded to obtain the EMG information for each class. The authors obtained feature vectors from EMG information based on mean absolute value, variance, waveform length, and zero crossing information. For each task, a Q-learning method was used, along with an artificial neural network as a policy representation, to tackle the classification problem. The authors stated that the RL method achieved better results than the ML approach for their dataset distribution. The RL-based classifier achieved accuracies of 97.51%, 98.73%, and 97.6% for the elbow angle, typing keys, and hand movement classification tasks, respectively. Other researchers proposed a neural reinforcement learning (NRL) method to classify finger movements related to typing keys using forearm EMGs [36]. The dataset was composed of 7 typing keys (‘h’, ‘k’, ‘o’, ‘shift’, ‘j’, ‘l’, and ‘;’) from 10 different subjects. In this work, four feature extraction methods were used: mean absolute value, variance, waveform Length, and zero crossing. The authors used the NRL classifier to characterize the keys using a trial-and-error approach. The proposed method achieved an accuracy of up to 99.01% for user-specific tasks (one model for each user data point) and up to 92.7% for the user-general model (one model for all user data). Finally, we presented an approach to classify and recognize EMGs using a Q-learning algorithm with an ANN as a policy representation of the agent in [30,37]. However, we only tested the simplest RL algorithm, and despite the obtained results being encouraging, other RL methods and agent configurations still need to be explored.

Considering the literature review presented above, the main contributions of the present work are listed as follows:We worked on our publicly available EMG dataset, “EMG-EPN-612”. This dataset contains EMG information of 612 users with data on five different hand gestures. We performed hyperparameter calibration to evaluate overfitting during the validation procedure of the proposed user-specific models.We trained classification and recognition methods based on EMG signals with two RL algorithms: Deep Q-network (DQN) and Double-DQN.For each algorithm, we proposed two different policy representations for the agent that are based on feed-forward artificial neural networks (ANNs) with and without a recurrent layer based on LSTM.

The rest of this article is structured as follows. In Section 2, the proposed HGR architecture that uses EMGs with either DQN or Double-DQN is presented, and each stage is explained in detail. The results are presented in Section 3. The discussion section is presented in Section 4. Finally, the conclusions are provided in Section 5.

## 2. Hand Gesture Recognition Architecture

In this section, we present the designed architecture to be used to solve the HGR problem based on EMG signals and DQN or Double-DQN (see Figure 1). The proposed architecture is conformed by a data acquisition stage, in which we explain the dataset. Then, we present the preprocessing stage, in which the EMG signal is segmented and filtered. In the feature extraction stage, the process to obtain the most relevant and non-redundant information is explained. In the classification stage, we explain how we used DQN and Double-DQN to solve the EMG signal classification problem. Finally, in the post-processing stage, we explain how to clean the obtained vector of labels to infer the recognition result. Next, we explain each stage in detail.

### 2.1. Data Acquisition

To build the dataset, we used EMG signals of six different hand gesture classes: fist, wave in, open, wave out, pinch, and relax (no gesture). To this end, we used the Myo armband sensor with a sampling rate frequency of 200 Hz for each of the eight channels. The EMG dataset used to develop this work is based on our online dataset, EMG-EPN-612, for which the acquisition protocol can be found in [38]. The EMG-EPN-612 dataset is presented in Figure 2. It can be seen that the dataset comprises 612 users, 306 of which are designated for training and validation for the process to find the best hyperparameter setup and the other 306 of which are designated for testing to evaluate overfitting and to calculate the final results. The data for each user in the training set comprise 300 hand gesture samples, 150 of which are designated for training and the other 150 of which are designated for validation. This division of samples is similar to the test set [39]. The online evaluation system, which is necessary to test our final models, can be accessed on the web page described in [40]. It is worth noting that we only used 20 users for training and validation (out of 306 available users) because the process of testing different user-specific models needed to find the best hyperparameters is computationally expensive and time-demanding. Once we found the best model and the best hyperparameter configuration for the 20 users using the validation set, we used that parameter configuration for training and testing based on the testing data to obtain the final results.

### 2.2. Preprocessing

To process each EMG sample, we used a segmentation procedure using a sliding window approach [1] with a window length of 300 and a stride of 40. Those values were selected empirically to achieve high classification and recognition accuracy. Then, we used a threshold of EMG energy to identify whether a current window observation would be processed for the classification stage or not. Thus, if the energy threshold of 18% was exceeded, then the current window observation was used; otherwise, this window information was considered a vector of zeros. Additionally, this preprocessing stage included a method for correction of the sensor orientation to make the proposed approach more robust against the EMG sensor’s rotation [39]. To this end, a set of synchronization EMG samples from the gesture “wave out” was used to determine the maximum energy channel, which was then used to correct the orientation. Thus, a new order of sensor pods was obtained based on the maximum average energy sensor predicted for the orientation correction algorithm. Next, the Myo bracelet sensor pods information was realigned with respect to the sensor with more energy.

### 2.3. Feature Extraction

The goal of the feature extraction stage is to collect non-redundant and relevant information from each EMG window to build a feature vector. Different methods can be used for this purpose, such as time- and/or frequency-domain approaches. In this work, five time-domain features were extracted for each EMG window observation. Then, if it surpassed the threshold of energy of 18%, we used the following feature extraction approaches: absolute envelope (AE), energy (E), root mean square (RMS), standard deviation (STD), and mean absolute value (MAV).

### 2.4. Classification of EMGS

The aim of this stage is to identify the category of an EMG among a set of categories with which the proposed algorithm was previously trained. To this end, we used two off-policy RL algorithms that are able to learn from online experience: Deep Q-Networks (DQNs) and Double-Deep Q-Networks (Double-DQNs). In this section, we explain the EMG sequential classification problem in detail, which can be considered a finite partially observable Markov decision process (POMDP).

#### 2.4.1. Q-Learning

The EMG sliding window classification can be considered a sequential decision-making issue, where the actions are the labels (gestures) used to estimate, and the states are the feature vectors that correspond to the window measurements. In this situation, we can determine how to calculate the best course of action for every state. In order to achieve this, we optimized the predicted sum of future rewards while taking into account the actions in the provided states, then implemented the best possible policy. If we take a policy (pi), we can define the value of the action (*a*) taken in the initial state (*s*) as
(1)$$Q_{\pi }\left(s,a\right)=E_{\pi }\left[R_{1}+\gamma R_{2}+\gamma^{2}R_{3}+\ldots +\gamma^{n-1}R_{n}|S_{0}=s,A_{0}=a\right]$$
where i=1,2,3,4,...,n; *n* denotes the number of states; $R_{i}$ denotes the rewards in each state; and γ
ϵ
[0,1] represents the discount factor, which controls the extent to which the agent’s process of learning is impacted by future rewards. The best state-action value function can now be written as $Q_{*}\left(s,a\right)={max}_{\pi }Q_{\pi }\left(s,a\right)$. By picking the highest-valued action in each state, the function $Q_{*}\left(s,a\right)$ can be used to derive an optimal policy. The off-policy temporal difference learning algorithm, Q-learning, can be used to learn estimates for the optimal state-action values [41,42]. Any finite Markov decision process (MDP) for which it is necessary to maximize the expected return from Equation (Equation 1) can have an optimal policy determined using the Q-learning approach. To this end, it is necessary to have an initial AC and for the agent to take a first action [30,41].

It is worth noting that we measure observations ($O_{t}$) instead of the information of the real state ($s_{t}$) from the environment because there is a probability of a mismatch between the set of EMG window observations that we obtained during the preprocessing stage and the set of feature vectors that we obtained during the feature extraction stage. Hence, it is possible for a feature vector to have the same value for two different window observations of an EMG sample. For this reason, we consider the HGR problem using EMGs as a partially observable Markov decision process (POMDP) [30].

When using the Q-learning algorithm, the Q-learning values can have different representations, such as lookup tables, polynomial functions, feed-forward neural networks, or even convolutional neural networks [41]. For the proposed approach, we used a continuous observation space (extracted features) and a discrete action space (predicted classes). Thus, we had to combine the Q-learning algorithm with a neural network representation as a function approximation method. In this context, the proposed Q-learning method can learn the value function $Q(O_{t},A_{t};\theta_{t})$.

Moreover, a critic can be used to represent the proposed agent in order to achieve outstanding outcomes for discrete action spaces [41]. A critic estimates the expected value of the whole long-term reward for a specific observation and action. Hence, after taking action $A_{t}$ in observation $O_{t}$ and earning reward $R_{t+1}$ in $O_{t+1}$, the Q-learning algorithm adjusts the neural network parameters ($\theta_{t}$), as explained in the following expression:(2)$$\theta_{t+1}=\theta_{t}+\alpha \left(Y_{t}^{Q}-Q\left(O_{t},A_{t};\theta_{t}\right)\right)\cdot \nabla_{\theta_{t}}Q\left(O_{t},A_{t};\theta_{t}\right)$$

The parameters that are used in the expression above represent the updated and the previous learned parameters ($\theta_{t+1}$ and $\theta_{t}$, respectively). The variable α represents the learning rate. The target function $Y_{t}^{Q}$ is defined as follows.
(3)$$Y_{t}^{Q}\equiv R_{t+1}+\gamma \cdot \underset{a}{max}\left[Q\left(O_{t+1},a;\theta_{t}\right)\right]$$
where $R_{t+1}$ represents the reward earned when going from observation $O_{t}$ to observation $O_{t+1}$ by taking action $A_{t}$. The optimal future Q value that is estimated is represented by the expression $\underset{a}{max}\left[Q\left(O_{t+1},a\right)\right]$, and the discount factor is denoted by γ.

#### 2.4.2. Deep Q-Networks (DQNs)

It is not realistic to describe the value function as a table with values for each possible combination for EMG classification issues, since the observation space is infinite. Instead, we employed a feed-forward multilayer artificial neural network (ANN) to develop the Deep Q-Network (DQN) agent representation. The DQN algorithm returns a set of action values for a certain observation ($O_{t}$), where $Q(O_{t},\cdot \;;\theta )$ denotes the parameters of the ANN [35,41,42]. The dimensions of the feature vector that represents an observation and the number of inputs in the network are the same. On the other side, the quantity of actions that the agent is capable of performing is used to determine the number of neurons in the output layer. For the DQN algorithm, there are two crucial properties to take into account. The target network ($Y_{t}^{DQN}$) used in Equation (Equation 4) contains parameters ($\theta^{-}$) that are updated regularly every τ steps from the online network in Equation (Equation 2), which has the parameters $\theta_{t}$. This is the first property. The parameters ($\theta^{-}$) are fixed for the remainder of the period until the subsequent update after τ steps. By doing this, correlations with the target are reduced [42,43].
(4)$$Y_{t}^{DQN}\equiv R_{t+1}+\gamma \cdot \underset{a}{max}\left[Q\left(O_{t+1},a,\theta_{t}^{-}\right)\right]$$

The use of experience replay to shuffle the data at random in order to eliminate correlations in the observation sequences is the second crucial property to take into account for the DQN algorithm. For this purpose, the vector $E_{t}=\left(O_{t},A_{t},R_{t},S_{t+1}\right)$, which represents the agent’s experience at time *t*, is saved in a set of stored data samples ($D=\{E_{1},E_{2},\cdots ,E_{T}\}$). Equations (Equation 2) and (Equation 4) are used to adjust the parameters of the ANN as it learns using mini batches of experience drawn at random from a pool of stored data samples (D) [43,44]. The performance of the DQN method is greatly enhanced by the use of the target network with parameters ($\theta^{-}$) and the experience replay approach [42,43].

#### 2.4.3. Double-Deep Q-Network (Double-DQN)

The max operator used to estimate the optimal future value of *Q* in standard Q-learning (Equation (Equation 3)) and DQN (Equation (Equation 4)) chooses and evaluates an action using the same values [42]. According to [42], this increased the probability of selecting overestimated values, which results in overoptimistic estimations of the values. To tackle this issue, we partially decoupled the selection from the evaluation using Double-Deep Q-learning (Double-DQN) [42]. In the original Double Q-learning algorithm (which is the basis for Double-DQN), there are two sets of weights, since two value functions are learned by assigning each experience randomly to update one of the two value functions. The first set of weights is aimed at determining the greedy policy; thus, the other calculates its value. However, in the Double-DQN algorithm, a candidate for the second value function is provided by the destination network. To accomplish this, the weights related to the target neural network ($\theta_{t}^{-}$) are used to evaluate the current greedy policy. This means that the action selection and evaluation for the case of Double-DQN are not fully decoupled as in the case of the Double Q-learning algorithm [42]. Thus, the Double-DQN algorithm evaluates the greedy policy using the online network from Equation (Equation 2) but working with the target network ($\theta_{t}^{-}$) to estimate its value. Thus, the update of the Double-DQN method is the same as for DQN ($\theta^{-}$ remains fixed until the next update after τ steps) but replacing the target ($Y_{t}^{DQN}$) with the following equation:(5)$$Y_{t}^{DoubleDQN}\equiv R_{t+1}+\gamma Q\left(O_{t+1},\underset{a}{argmax}Q\left(O_{t+1},a;\theta_{t}\right);\theta_{t}^{-}\right)$$

#### 2.4.4. Deep Recurrent Q-Networks with LSTM

For certain applications and dataset distributions, it has been observed that for problems based on POMDP, the use of deep recurrent Q-networks can achieve better results than DQN methods [45,46]. However, this does not apply to all applications because using LSTM can also confuse the agent [46]. In this work, we also implemented a deep recurrent Q-network with LSTM to evaluate whether, for our application, the performance would increase or decreases using an LSTM layer for both the DQN and Double-DQN algorithms.

#### 2.4.5. POMDP for EMG Classification

The partially observable Markov decision process (POMDP) proposed in this work uses the DQN or Double-DQN-based algorithm to learn an optimal policy to classify and recognize EMGs. An illustration of the agent–environment interaction using the Deep Q-Network algorithm for EMG classification and recognition is presented in Figure 3. We explain each of the elements of Figure 3 in detail below.

Agent: To create an RL-based agent, we need to combine an RL algorithm with a policy representation. The main idea is that a policy that maximizes the total sum of rewards using DQN or Double-DQN RL algorithms can be learned by the agent. The agent in our work is composed of an ANN, which returns the value of the actions that the agent can take at any observation of the environment. For the agent representation, we compared two different architectures: a neural network and a neural network with an LSTM layer, as illustrated in Figure 4. Our agent learns to classify each window observation of an EMG-based hand gesture sample, and each sample represents an episode in which an agent interacts with the environment. For a given observation, our agent returns an estimate of the expected value of the cumulative long-term reward for the proposed task.

Observation: We define the observation ($O_{t}$) as the feature vector resulting from the feature extraction process, which is composed of information about the AE, STD, RMS, MAV, and E for each window observation of the EMG. The number of the last observation obtained from the sliding window for an EMG sample is defined as *N*; thus, the last observation that occurs at the end of an episode is denoted by $O_{N}$.

Action: An agent can take an action ($A_{t}$) at the current observation ($O_{t}$) to reach observation $O_{t+1}$ and receive reward $R_{t+1}$. We define an agent’s action as the label assigned to each window observation of the EMG signal. The labels used for this work are fist, wave in, open, wave out, pinch, and relax (no gesture).

Environment: The agent takes an action within a defined environment, which returns a reward and the next state or observation of the agent. In this case, we define the environment as the combination of the sliding window and the ground truth of each EMG signal. The sliding window provides a different EMG feature vector at each time step, and the EMG ground truth provides the true category label for each EMG signal sample.

Reward: The reward is a numeric value that the agent receives depending on its performance when interacting with the environment. An illustration of the rewards that the agent obtains is presented in Figure 5. In this work, we used classification rewards, which can be positive ($R_{t}=$ +1) each time the agent correctly predicts a category label (action) based on a feature vector (observation). If the agent predicts a window label category incorrectly, it receives $R_{t}=$ −1. We also used a recognition reward, which compares the vector of predicted labels with the vector of true labels (ground truth). As can be observed in Figure 5, if the overlapping factor between those vectors is more than 75%, the agent receives a reward ($R_{t}=$ +10); oitherwise, the agent is penalized with $R_{t}=$ −10.

### 2.5. Post Processing

Once an EMG sample is processed, we use post processing to eliminate spurious labels from the prediction vector of categories. It is worth mentioning that the post-processing stage considerably improves the accuracy of the HGR system. Most works found in the literature use filtering, majority voting, mode, heuristics, and threshold approaches [1,30]. Although we tried several methods to remove spurious labels, the method with which we obtained the best results was majority voting. The basic idea of this method is to count how many predictions of each category label were made at the end of an episode. Then, the category label that is repeated the most times replaces the others, but only where there is muscle activity. In the parts of the signal where there is no muscular activity, the relax gesture (no gesture) is inserted.

## 3. Results

We first present the validation results for the user-specific HGR model, which were necessary to find the best possible hyperparameters by testing on EMG data from 20 users. Then, we present the final test results for 306 users with the best hyperparameters found during the validation procedure. The results of the validation and tests are explained below.

### 3.1. Validation Results

For the validation results, we trained and tested different models based on agents that use neural networks as policy representations (with and without LSTM) with the DQN and Double-DQN algorithms that we presented previously in Section 2.4. We summarize the seven models that we used for validation in Table 1.

For each model, we tested different hyperparameters such as the learning rate and the update frequency. The hyperparameter values for our experiments are summarized in Table 2. It is worth mentioning that we carried out a large number of experiments, changing various hyperparameters. However, to show the validation results, we only changed the learning rate and the target update frequency. The rest of the parameters that were obtained and which were the best remain fixed.

A training sample illustration of the average reward versus the episode number (number of epochs) for the DQN-based model is illustrated in Figure 6. The convergence of the estimate of the discounted long-term reward at the start of each episode(yellow line) to the episode reward (blue line) and the average reward of the last five steps (red line) can be observed.

A summary of the classification and recognition accuracy results for 20 users of the proposed user-specific HGR model can be observed in Figure 7, Figure 8 and Figure 9. In Figure 7, we present a comparison of DQN and Double-DQN with update frequency =1. It can be observed that α= 0.0003 for the DQN-based model achieves the best results: are 85.7%±3.8% and 84.1%±12% for classification and recognition accuracy, respectively. In Figure 8, we present the results corresponding to an increase in the update frequency for the Double-DQN case. It can be observed that α=0.0003 for the Double-DQN-based model with update frequency =5 achieved the best results: 83.6%±3.1% and 79.9%±8.5% for classification and recognition accuracy, respectively. Finally, in Figure 9, we present the results for the DQN and Double-DQN models with an LSTM layer for the agent representation of those models. It can be observed that $\alpha =0.09\times 10^{-3}$ for the Double-DQN based model achieved the best results for this model: 51.6%±17.4% and 26.6%±21.7% for classification and recognition accuracy, respectively. This implies that adding an LSTM layer drastically decreases the performance of the agent in classifying and recognizing EMGs using the proposed models. In conclusion, the best model found during the validation procedure was obtained from the results illustrated in Figure 7 with α=0.0003 for the DQN-based model, which will be used for the test procedure. In addition, numerical results for the best result of each model are presented in Appendix A in Table A1.

### 3.2. Testing Results

Based on the best hyperparameters found during the validation procedure, as reported in Section 3.1 (α=0.03 for the DQN model and the parameters from Table 2), we used the test set composed of 306 users to evaluate such user-specific models. This procedure helped us to evaluate our models with different data and analyze overfitting. The test accuracy results for 306 users in the testing set were 90.37%±10.7% and 82.52%±10.9% for classification and recognition accuracy, respectively. We present a summary of the results of the testing procedure for the 306 users with the best hyperparameters in Table 3. It should be noted that there is barely a difference between the testing and validation accuracy results. Therefore, we can infer that our models are robust, since overfitting for the proposed distribution of the dataset does not affect their performance. Moreover, it can be seen that the test accuracy results outperform the validation results because of the dataset distribution, since for validation, we used only 20 users, whereas for testing, we used 306 users. A confusion matrix was used to illustrate the classification results on the test set, as shown in Figure 10, which allows us to observe, in detail, the results for each hand gesture. Finally, we present an illustration with the classification and recognition results for each of the 306 test users in Figure 11. The average processing time for each sliding window observation is 32.2 ms.

## 4. Discussion

To date, conventional methods for designing HGR systems have mainly focused on the application of supervised machine learning or deep learning algorithms, including memory-using models such as LSTM and CNN [16]. However, the use of RL to develop HGR systems is only beginning to be investigated. The present work compares different variants of RL-based algorithms, i.e., DQN and Double-DQN, with and without LSTM layers. The proposed approaches were implemented in the largest known dataset of EMGs, EMG-EPN-612. The proposed approaches were compared in terms of classification and recognition accuracy, providing valuable information for the development of HGR systems based on EMG signals.

The approaches used in this work are intended to contribute to the development of HGR systems based on EMG signals using RL techniques. The best model accuracy for the validation dataset achieved accuracies of 85.7%±3.8% and 84.1%±12% for classification and recognition, respectively, with the DQN model. Moreover, the accuracy of such a model for the validation dataset reached up to 90.37%±10.7% and 82.52%±10.9% for classification and recognition, respectively. These results demonstrate the effectiveness of the proposed algorithms. To achieved such results, extensive experiments were carried out, which included hyperparameter calibration and comparison of each RL-based model in terms of classification and recognition accuracy. The proposed DQN algorithm is also very useful for learning online based on the experiences of each user, which is something that only RL-based algorithms can achieve. This can help improve the models by allowing for readjustment of the agent policy representation weights to reduce intraperson and interperson variability and make each model fit the needs of each user.

It is important to mention that to fairly evaluate of the proposed algorithm, it is important to compare it with other methods that use the same dataset, i.e., EMG-EPN-612. Although we found an extensive number of works in the literature related to EMG signals, a comparison with these studies is not objective because such a comparison depends on the data distribution, quantity, and quality of the data selected for training, as well as the quantity and type of the selected hand gestures (some gestures are easier to recognize than others). For example, in the work presented in [47], the authors used an RL method based on a CNN agent representation and DQN algorithm with the Myo armband sensor. This approach achieved a classification accuracy of 98.33%. However, the authors did not use EMG signals but quaternions. Additionally, the amount of data used in that work was only 90 samples of hand gesture data. In the present work, we used the EMG-EPN-612 dataset, which comprises 300 samples of gestures for each of 612 different users (a total of 183,600 gestures). This makes the results obtained statistically relevant, since we tested a large amount of data. Other works, such as [21], used a CNN for the classification of 50 hand movements from 67 users and 11 transradial amputees, achieving an accuracy of up to 90%. However, the methods they used did consider the same amount of data as we used in the current work. Moreover, they used a CNN approach based on supervised deep learning (DL) methods and not reinforcement learning (RL). In summary, reinforcement learning methods are just beginning to be investigated for applications in HGR systems based on EMG signals, which makes objective comparison with other methods difficult at this stage.

It is worth noting that RL algorithms are able to learn from sequential problems, which can be easily applied to the problem of a sliding window-based HGR system. In the present work, we were able to demonstrate that the use of RL algorithms such as DQN is efficient for developing HGR systems. However, we found that using an additional LSTM layer for the neural network to build the agent’s policy representation drastically decreased the accuracy of the results (down to 49.8%). This is consistent with previous findings demonstrating that, depending on the application, adding LSTM layers does not always lead to the development of better DQN approach policies [46].

In this work, preprocessing, feature extraction, classification, and post-processing stages were selected based on extensive experiments that allowed us to find the best fit for our dataset distribution. For example, for the feature extraction stage, we used absolute envelope (AE), energy (E), root mean square (RMS), standard deviation (STD), and mean absolute value (MAV). These characteristics were selected based on the results of several experiments that we carried out; we chose the stages and configurations that allowed us to achieve the greatest possible classification and recognition results. For example, for the post-processing stage, based on observation of the resulting predicted signals, we observed that filtering was necessary. To this end, we tried several techniques, such as mode, heuristics, threshold, and majority voting, with which we obtained the best results.

One of the important contributions of this work to the HGR systems development area is that the proposed algorithm learns based on sequential information, which is inherent in EMG signal recognition problems. Moreover, current commercial systems suffer from the problem of intraperson and interperson variability. This means that the performance of existing commercial sensors can vary from person to person and does not adapt to the needs of each user. In this work, we used RL algorithms based on DQN and Double-DQN, which are very useful for online learn depending on the experiences of each user, providing HGR models an adaptation capacity that conventional methods do not have. The proposed algorithm can help improve the models by allowing them to readjust online to reduce intraperson and interperson variability.

In future works, we will deepen the use of RL for applications of HGR systems. In particular, we will explore the use of new RL agents and different representations of their policy, such as CNN-based agents. Likewise, it is important to continue exploring the use of different data types to implement an HGR system, for example, the use of more sensors on the arm; the use of additional sensors, such as inertial measurement units (IMUs); the use of flex gloves; and/or the combination of all these sensor signals. In the future, we will carry out a qualitative evaluation of the user experience when using RL techniques so that the models adjust to the characteristics of each user, thereby reducing inter- and intrapersonal variability. Finally, in future works, we will work with a large database with different sensor positions and wrist and elbow angles to test the proposed RL method for HGR applications. This will contribute to the creation of a model that is general and robust, that is, that can be used for all users and that it is not affected by changes regarding the sensor position and angle on the wrist and elbow.

## 5. Conclusions

In this article, we developed an HGR architecture based on two reinforcement learning algorithms for the classification stage: DQN and Double-DQN. For each algorithm, we tested and compared different agent representations, such as a feed-forward neural network with and without an LSTM layer to test the agent performance. The developed models were tested for a validation set of 20 users. In this context, we demonstrate that the best classification accuracy results can reach up to 85.7%±3.8%, with a recognition accuracy of 84.1%±12% with the DQN model with α=0.03. Based on such a model, we evaluated the classification and recognition accuracy with the test set of the EMG-EPN-612 dataset comprising 306 users. During testing, the classification accuracy can reach up to 90.37%±10.7%, while the recognition accuracy reached up to 82.52%±10.9% for the DQN model with α=0.03. The results obtained in this work show that DQN (without LSTM) was the best method, allowing an agent to learn a policy to classify and recognize gestures based on EMG signals. In addition, the results show that the use of Double-DQN does not improve the classification and the recognition results for the proposed POMDP. Finally, it was also observed that the use of an LSTM layer dramatically affects the performance of the algorithms for this application and data distribution. In this case, it is concluded that the simplest model based on DQN achieves better results than more complex modifications. Future work will focus on testing other feature extraction methods and reinforcement learning algorithms to evaluate the proposed POMDP.

## Figures and Tables

**Figure 1 sensors-23-03905-f001:**
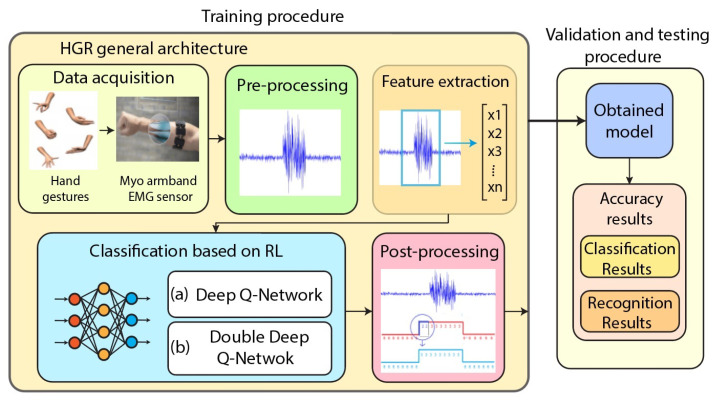
Hand gesture recognition architecture based on EMGs and RL. The classification stage can use either DQN or Double-DQN.

**Figure 2 sensors-23-03905-f002:**
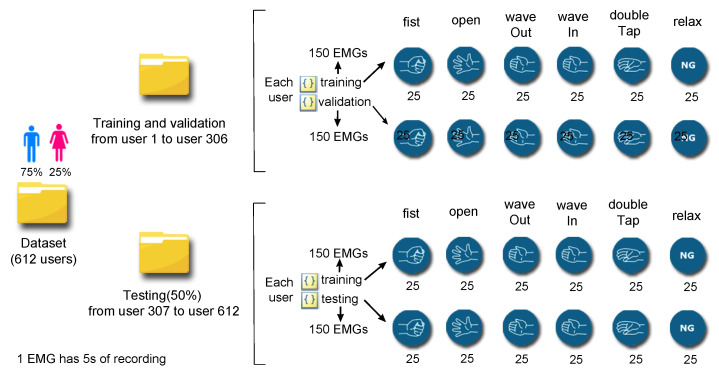
EMG-EPN-612 dataset. One model was trained for each user data point (user-specific approach).

**Figure 3 sensors-23-03905-f003:**
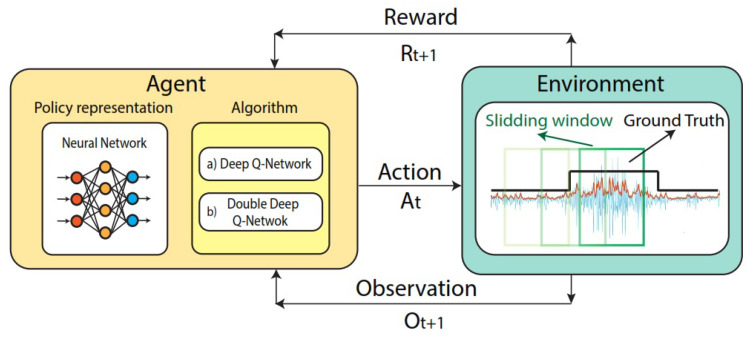
Scheme of the interaction between the Deep Q-Network agent representation and the proposed environment for EMG classification.

**Figure 4 sensors-23-03905-f004:**
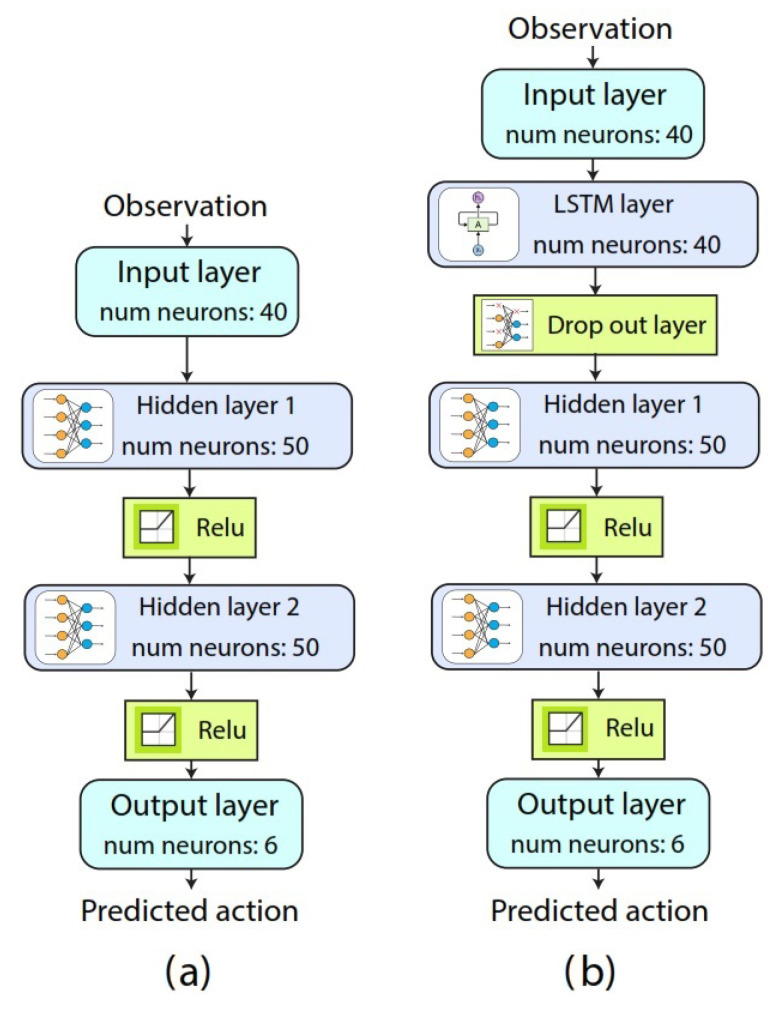
Representations proposed for the agents. (**a**) A neural network representation. (**b**) A Neural network representation with an LSTM recurrent layer.

**Figure 5 sensors-23-03905-f005:**
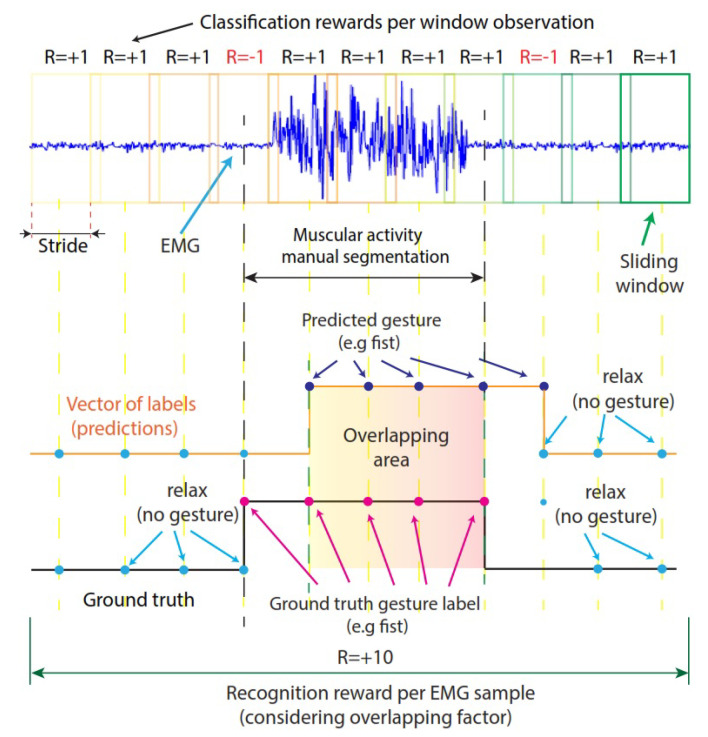
Rewards that the agent obtains when it interacts with the environment.

**Figure 6 sensors-23-03905-f006:**
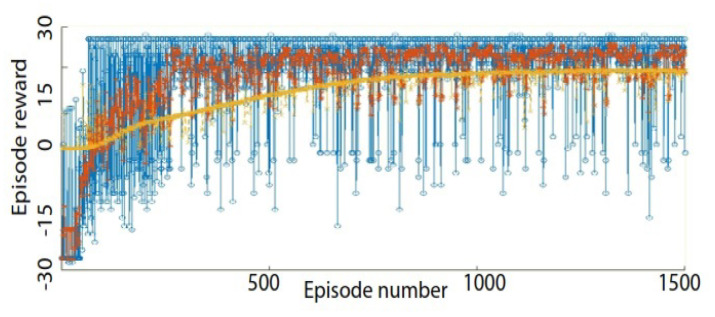
Episode rewards versus episode number. User-specific HGR training sample results for User 1 with DQN. We can observe the convergence of the estimate of the discounted long-term reward at the start of each episode (yellow line) to the current episode reward (blue line) and average reward of the last five steps (red line).

**Figure 7 sensors-23-03905-f007:**
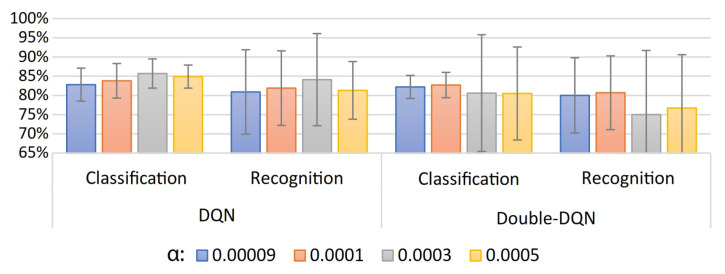
User-specific HGR model classification and recognition accuracy results for 20 users with different values of learning rate (α) for both DQN and Double-DQN.

**Figure 8 sensors-23-03905-f008:**
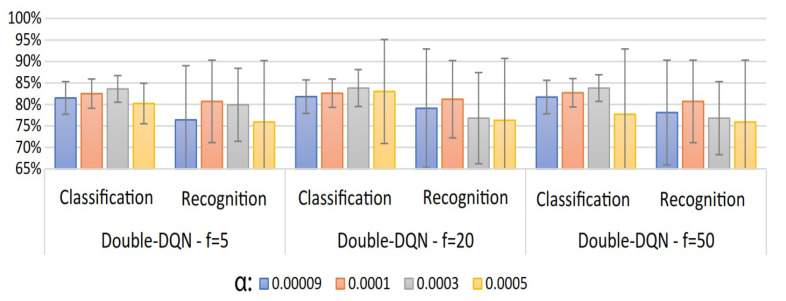
User-specific HGR model classification and recognition accuracy results for 20 users with different values of learning rate (α) and update frequencies for Double-DQN.

**Figure 9 sensors-23-03905-f009:**
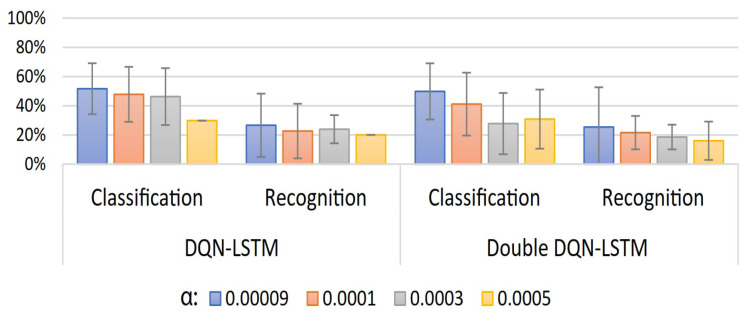
User-specific HGR model classification and recognition accuracy results for 20 users with different values of learning rate (α) for both DQN and Double-DQN with LSTM.

**Figure 10 sensors-23-03905-f010:**
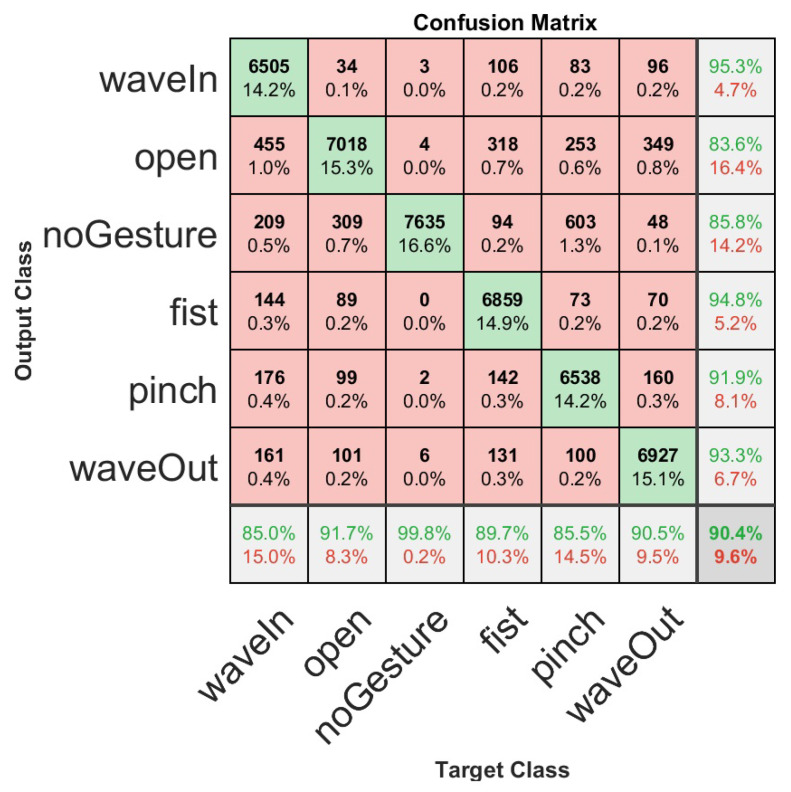
User-specific HGR model confusion matrix for 306 users of the test set with the best hyperparameter configuration. Each hand gesture class result can be observed in detail.

**Figure 11 sensors-23-03905-f011:**
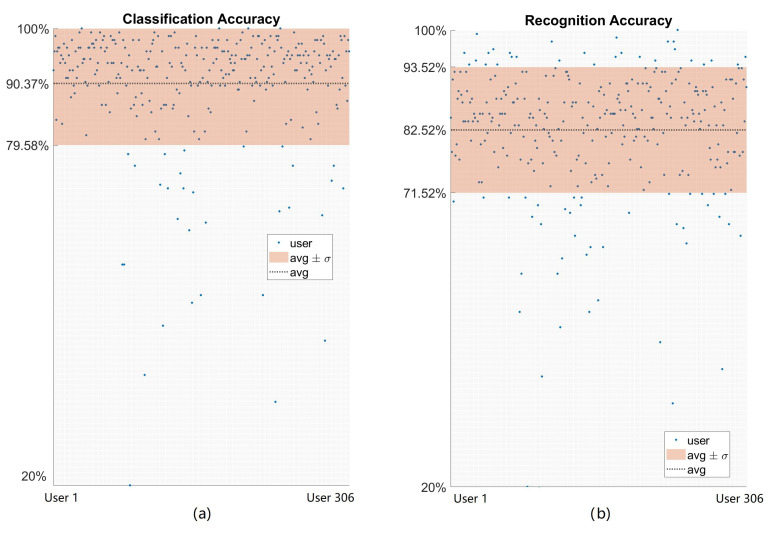
User-specific HGR model results for each of the 306 users of test set with the best hyperparameter configuration: (**a**) classification results; (**b**) recognition results.

**Table 1 sensors-23-03905-t001:** Proposed models for experiments.

Model	Policy Representation of the Model	Algorithm
1	Neural network (Figure 4a)	DQN
2		Double-DQN frec update = 1
3		Double-DQN frec update = 5
4		Double-DQN frec update = 20
5		Double-DQN frec update = 50
6	Neural network + LSTM layer (Figure 4b)	DQN
7		Double-DQN frec update = 50

**Table 2 sensors-23-03905-t002:** Hyperparameters used for the tuning procedure during validation.

Hyperparameter Name	Hyperparameter Values
Number of neurons per layer	40, 50, 50, and 6 for the input layer, hidden layer 1, hidden layer 2, and output layer, respectively
LSTM layer (only for models with a recurrent layer)	27 neurons and a drop-out layer
Activation function between layers	ReLU
Target smooth factor	$5\times 10^{-3}$
Target update frequency (for Double-DQN)	1, 20, and 50
Experience buffer length	$1\times 10^{6}$
Learn rate (α)	$0.09\times 10^{-3}$, $0.1\times 10^{-3}$, $0.3\times 10^{-3}$, and $0.5\times 10^{-3}$
Epsilon initial value	1
Greedy epsilon decay	$5\times 10^{-3}$
Discount factor	0.99
Training set replay per user	15 times
Sliding window size	300 points
Stride size	40 points
Mini batch size	64
Optimizer	Adam
Gradient decay factor	0.9
L2 regularization factor	0.0001

**Table 3 sensors-23-03905-t003:** User-specific validation and test results of the HGR best model.

Validation Set (20 Users)		Test Set (306 Users)	
Classification	Recognition	Classification	Recognition
85.7%±3.8%	84.1%±12%	90.37%±10.9%	82.52%±10.9%

## Data Availability

Publicly available datasets were analyzed in this study. This data can be found here: https://laboratorio-ia.epn.edu.ec/es/recursos/dataset/2020_emg_dataset_612 (accessed on 5 April 2023).

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
