# Peer review of "Recognition of Hand Gestures Based on EMG Signals with Deep and Double-Deep Q-Networks"

_sensors, 2023, doi:10.3390/s23083905_

Round 1

Reviewer 1 Report

1.      The paper is not innovative enough, many similar studies have been done on EMG signals for gesture recognition. The authors need to think about what new problems can be solved by reinforcement learning (RL) techniques based on previous research.

2.      Simply replacing traditional machine learning(ML) with RL classification model is not enough to support the innovative points of the paper, and the accuracy of the results is not satisfactory.

Here are some additional specific comments,

1.The paper focuses on the application of reinforcement learning to the gesture classification of EMG signals, which is not innovative enough because it simply replaces the traditional machine learning method with reinforcement learning.

2.The proposed reinforcement learning approach in the paper does not address the new problem of sEMG for gesture recognition.

3.The authors need to further condense what new problems can be solved by reinforcement learning for sEMG for gesture recognition. There is already a lot of literature on deep learning for sEMG signal gesture classification.

4.Figure 6 lines are very messy, can not show the intended meaning.

Author Response

Dear reviewer,

Thank you very much for your comments and suggestions. These have been very useful to improve the performance of this article. The comments and suggestions can be found in the attached file.

Reviewer 2 Report

In this paper, the authors have proposed a user-specific hand-gesture recognition model by using a deep reinforcement learning (RL) approach. Their RL model learns to characterize EMG signals from five different hand gestures by using Deep Q-network (DQN) and Double Deep Q-Network (Double-DQN) algorithms. Additionally, they have used feed-forward artificial neural network (ANN) for the representation of the policy, as well as a Long-short term memory (LSTM) layer to analyze and compare their performance. They have used the public dataset of EMG-EPN-612 through which they have concluded that the performance of the model with but DQN without (LSTM) was the best obtaining up to 90.37%± 10.7% and 82.52%± 10.9% classification and recognition accuracy respectively. The have finally concluded that their results demonstrate the capability of RL models in classification and recognition problems based on EMG signals.

While the reviewer founds the work interesting with sound scientific approach, the following points should be addressed to improve the manuscript: 

- What is (are) the advantage(s) of using RL over unsupervised approaches? 

- How would you compare the performance of an RL model with a Deep Learning (DL) model (ex. CNN)? The CNNs have shown promising performance. For example: 

[Gopal, P.; Gesta, A.; Mohebbi, A. A Systematic Study on Electromyography-Based Hand Gesture Recognition for Assistive Robots Using Deep Learning and Machine Learning Models. Sensors 2022, 22, 3650. https://doi.org/10.3390/s22103650]

- RL is used here as it does not require labels. The same holds for Unsupervised learning approaches. 

- RL is usually used to learn a policy at the same time that the experience is done and actions are generated. If you already have a dataset, why using RL and not unsupervised DL or ML ? 

- You should clearly justify your choice of RL and after that your choices of the modules/algorithms used. 

- The font size for the equations should be increased. 

- Figure 1 does not contain any process or notion of RL, yet it is mentioned in the caption. 

- Figure 6 is unclear. The resolution should be increased while labels should be used with different colors for each plot. It is totally unclear what the figure is showing and referring to. 

- Based on the figures 7-9, the variations in results are high. How did you test the significance of the results and how did you conclude based on these results? 

Author Response

(The authors gave the same response as above.)

Reviewer 3 Report

The paper presents a RL technique for hand gesture recognition based on EMG. The results are interesting and have great potentials for application. For accepting the manuscript, I have the following main comments:

- only EMG and IMU based sensors are discussed, but there is a lot of literature around ultrasound-based gesture recognition as well. I suggest to include some references along this line of research as well in the introduction (e.g., doi: 10.1109/TNSRE.2022.3205026)

- one third of all cited references are self-citations, I believe it's too much

- comparison to state of the art requires improvement. Performance metrics (e.g. accuracy) should be compared to what has been reported in similar works in literature. Examples of works include (but should not be limited to): 10.1109/EMBC48229.2022.9871898, 10.3389/fnbot.2016.00009,10.1109/ICOIN.2018.8343257

Author Response

(The authors gave the same response as above.)

Round 2

Reviewer 1 Report

1.Because EMG based HRG system has a very mature commercial application, their online gesture recognition has achieved good results. I don't think it is innovative enough to publish in Sensors with using RL method for traditional EMG based HRG system.

2.The author explains in the Discussion that EMG-EPN-612 is a very large data set, including 300 samples of gestures for each of the 612 different users (a total of 183.600 gestures). Different subjects may have sensor position drift during the experiment, resulting in poor signal consistency. How can the author solve this problem?

3.The author emphasizes that RL has good learning ability for HGR system, and even has better performance than deep learning. Simple EMG based HRG has done a lot of related research, the author can try to use RL for HRG under different wrist or elbow joint angles, and explore whether RL can overcome the interference caused by the change of wrist and elbow joint angles.

Author Response

Dear reviewer,

Thank you for your time and valuable responses. We have made a great effort to try to respond and solve as clearly and concisely as possible to all of your suggestions and questions.

You can find the response letter attached below.

Best regards

Reviewer 2 Report

Although the authors did not provide a rebuttal letter with clear answers to my question, I can observe that they have applied most of my suggestions. 

Author Response

(The authors gave the same response as above.)
